# Reduced Environmental Dose Rates Are Responsible for the Increased Susceptibility to Radiation-Induced DNA Damage in Larval Neuroblasts of *Drosophila* Grown inside the LNGS Underground Laboratory

**DOI:** 10.3390/ijms23105472

**Published:** 2022-05-13

**Authors:** Antonella Porrazzo, Giuseppe Esposito, Daniela Grifoni, Giovanni Cenci, Patrizia Morciano, Maria Antonella Tabocchini

**Affiliations:** 1Dipartimento di Biologia e Biotecnologie “C. Darwin”, Sapienza Università di Roma, 00185 Rome, Italy; antonella.porrazzo@uniroma1.it (A.P.); gianni.cenci@uniroma1.it (G.C.); 2Centro Nazionale per le Tecnologie Innovative in Sanità Pubblica (TISP), Istituto Superiore di Sanità (ISS), 00161 Rome, Italy; antonella.tabocchini@iss.it; 3Istituto Nazionale di Fisica Nucleare (INFN), Sezione Roma 1, 00185 Rome, Italy; 4Dipartimento di Medicina Clinica, Sanità Pubblica, Scienze Della Vita e Dell’ambiente, Università Dell’aquila, 67100 L’Aquila, Italy; daniela.grifoni@univaq.it; 5Fondazione Cenci Bolognetti, Istituto Pasteur, 00185 Rome, Italy; 6Laboratori Nazionali del Gran Sasso (LNGS), INFN, Assergi, 67100 L’Aquila, Italy

**Keywords:** environmental radiation, deep underground laboratory, *Drosophila melanogaster*, chromosome aberrations, dose rate effect

## Abstract

A large amount of evidence from radiobiology studies carried out in Deep Underground Laboratories support the view that environmental radiation may trigger biological mechanisms that enable both simple and complex organisms to cope with genotoxic stress. In line with this, here we show that the reduced radiation background of the LNGS underground laboratory renders *Drosophila* neuroblasts more sensitive to ionizing radiation-induced (but not to spontaneous) DNA breaks compared to fruit flies kept at the external reference laboratory. Interestingly, we demonstrate that the ionizing radiation sensitivity of flies kept at the LNGS underground laboratory is rescued by increasing the underground gamma dose rate to levels comparable to the low-LET reference one. This finding provides the first direct evidence that the modulation of the DNA damage response in a complex multicellular organism is indeed dependent on the environmental dose rate.

## 1. Introduction

Natural ionizing radiation (IR) has been regarded as a crucial factor in the evolution of life forms on earth since the first cell came into being about 4 billion years ago. It has been postulated that the ability of present-day organisms to cope with radiation-induced DNA damage could in part result from the adaptation of their ancestors, who experienced many fluctuations of environmental radiation exposure (i.e., very high radiation exposure) during primeval times [1]. The natural radiation, whose dose rate currently varies in the range of 10^−7^–10^−5^ Gy/h, results from a combination of cosmic, terrestrial, and internal sources of different forms of energy and electrically charged particles, generally referred to as background radiation. It is noteworthy that background radiation also shows large geographical variations. Since the exposure of organisms, including humans, to background radiation is unavoidable, understanding its role is important to address basic questions about life’s evolution on earth and the health effects of low-dose ionizing radiation exposure, a relevant issue in radiation protection. In this respect, controlled experiments with model organisms, conducted in parallel in underground laboratories where the radiation background is largely reduced, and in reference conditions at a natural background radiation, could provide useful insight into the overall role of natural radiation. Among the various locations that hosted experiments in conditions of deprived background radiation, the Deep Underground Laboratories (DULs) stand out for their unique shielding characteristics that make it possible to reduce the total radiation background. These research infrastructures are built under a rock overburden greater than about 1000 m of water equivalent (m.w.e), originally created to host particle, astroparticle, and/or nuclear physics experiments (i.e., the search for neutrino interaction, proton decay, matter particles), which would require stringent experimental conditions of strongly reduced cosmic ray particle interference [2,3]. Being among the most efficient places for experimental isolation from radiation, some DULs have become multidisciplinary sites hosting important studies in fields such as geology, geophysics, the climate, environmental, and space sciences, technology/instrumentation development, and radiobiology. So far, all biological studies carried out in DULs strongly indicate that the deprivation of natural background radiation affects, although to varied degrees, the growth and transcription profiles of bacteria as well as of unicellular and multicellular eukaryotes [1,4,5,6,7]. Moreover, several experiments indicate that cell cultures kept in this strongly reduced background show higher susceptibility to subsequent radiation-induced DNA damage than parallel cultures kept in external radiation background [8,9,10]. These observations challenge the Linear No-Threshold (LNT) model, which posits that all radiation exposure is always considered harmful with no threshold point and that risks increases linearly with the dose [1,11].

The Gran Sasso National Laboratory (LNGS) at Assergi (L’Aquila, Italy) is one of the largest DULs in the world, hosting the largest number of biological experiments performed to date. Due to the 1400-m coverage of dolomitic rocks poor in uranium and thorium, at LNGS the flux of cosmic rays is considered negligible as it is reduced by approximately 6 orders of magnitude, while the neutron flux is 1000 times reduced with respect to the external environment [12,13]. The setting up of two dedicated facilities, namely, PULEX and COSMIC SILENCE for cell culture and animal housing, respectively, allowed researchers to gather several pieces of evidence on the influence of the deprivation of a natural level of environmental radiation in different model systems [12,14]. In particular, a few years ago, we launched the FLYINGLOW project that aimed to determine whether the LNGS’s underground environment could affect the growth and development of the well-established model organism, *Drosophila melanogaster*. Our results indicated that the reduced background radiation affects the lifespan and fertility of adult flies. Providing the first evidence of the influence of radiation background in a complex organism [3]. Since then, other organisms such as fishes [15] and nematodes [6] have been demonstrated to promptly modify their physiological responses following changes in the radiation background. Collectively, these observations suggest that the stimulation of defense mechanisms against stress triggered by the natural background radiation is an evolutionarily conserved phenomenon. However, to date, no direct evidence has been provided about the involvement of environmental radiation in determining these different responses. It should be considered that. not only underground but also in an external environment, a small number of cells within an organism interact with radiation every week [16]. Nevertheless, even if few cells are directly affected by radiation, cell–cell communication mechanisms may be able to propagate signals to other (neighboring) cells and thus amplify the total number of perturbed cells [17]. Moreover, the low- and high-LET components of the environmental background could have a different weight. Thus, dedicated experiments are needed to clarify this issue.

In different DULs, various components contribute to the radiation field. Generally, whereas both the high-LET (e.g., neutrons) and low-LET (e.g., muons) component of cosmic rays can be reduced by several orders of magnitude, the low-LET terrestrial gamma rays and the products deriving from ^222^Rn decay contribute significantly to the overall dose/dose rate. The characterization of the radiation field remains a crucial point for the interpretation of the biological effects, and dosimetric measurements should be constantly carried out not only underground but also in the reference laboratory. Furthermore, the modulation of the radiation field inside the DULs through the implementation of devices that could modulate some of these components could provide insights into the mechanisms underpinning the observed biological effects [12,18].

Here, we show for the first time that the reduced natural background radiation (herein Low Radiation Environment, LRE) at LNGS affects the response to radiation-induced DNA damage in the complex eukaryotic system *Drosophila melanogaster.* In particular, we show that neuroblasts from wild-type flies grown for one and five generations at the LNGS LRE exhibit a frequency of chromosome breaks (CBs) induced by acute exposure to ionizing radiation higher than that observed in cells from flies kept at the external (herein Reference Radiation Environment, RRE) LNGS facility. Moreover, we show that increasing the low-LET component by means of a Marinelli beaker designed ad hoc and filled with tuff, a natural gamma emitter building material, rescues the IR sensitivity in cells from flies kept at LRE. This finding provides the first evidence that the modulation of the DNA response observed in *Drosophila melanogaster* neuroblasts is indeed radiation-dose-rate-dependent and that a minimum dose rate level (comparable to the environmental one) is required to trigger efficient DNA damage response mechanisms.

## 2. Results and Discussion

In the present paper. we set out to investigate if reduced background radiation affects the response to radiation-induced DNA damage, expressed as CBs/cell, induced by an acute dose of gamma rays in *Drosophila melanogaster* wild-type mitotic cells. To this purpose, we raised *Oregon-R* (wild-type) flies at LRE and in parallel at RRE for one and five generations. Third instar larvae were collected from generation 1 and generation 5 and irradiated with an acute challenging gamma dose of 10 Gy; four hours later, larval neuroblasts were analyzed for CB frequency. The average number of total CBs/cell was calculated by measuring the ratio of the total number of chromatid deletions (CDs, scored as a single event; see also Figure 1b) and isochromosome breaks (ISOs or chromosome deletions, scored as two events; see also Figure 1c) to the total number of metaphases (Figure 1a). Whereas the CB frequency induced by the challenging dose in the RRE larvae was ~0.6, larvae from LRE flies exhibited a ~2-fold increase in the CB frequency (Figure 2a). This finding suggests that the reduced natural background affects the response to radiation-induced chromosome damage. Interestingly, we found that the CB frequency from the generation 5 LRE was indistinguishable from that of the generation 1 LRE, indicating that the *Drosophila* cell sensitivity is not caused by the occurrence of spontaneous dominant mutations but rather represents an early response to the switch from a normal to reduced radiation background. Similar effects were also previously described in flies kept at LRE for several generations, which showed reduced fertility [13]. It is also interesting to note that the deprivation of the radiation background at LRE did not render the DNA repair and telomere capping mutant flies [19,20,21,22] more sensitive to the chromatin burden induced by endogenous genome instability compared to RRE (Figure 3). This confirms the idea that extremely low doses, such as the environmental ones, increase the ability of cells and organisms to cope with exogenous damage through a stress response, not a damage response. The mechanisms involved in these phenomena need to be further investigated, although we can envisage that epigenetic mechanisms could play a relevant role [1].

Since the beginning of our underground biology activity, we applied an experimental approach to minimize all the possible differences between the underground and reference environments except radiation, and collected convincing indications that the different behavior depends on the different radiation field in LRE and RRE [9,13,24]. It is clear that the percentage of cells in the biological system that interacts with radiation at RRE and LRE is very low, and therefore we must assume a strong involvement of cellular communication effects. These effects have been clearly demonstrated to occur at low doses above ground [25]. In principle, the observed difference could also be linked to some other factors difficult to control (i.e., differences in air quality, vibrations related to nearby installations, the presence of sunlight at RRE and not at LRE, etc.) as proposed by Zarubin et al. [7]. We used incubators to avoid at least some of the proposed confounding factors and tried to get more insight on the involvement of the environmental radiation on the modulation of the radiation-induced DNA damage, directly investigating the role of specific components of the environmental radiation field on the response of *Drosophila* larval neuroblasts [12]. In line with the aims of our recently launched RENOIR (Radiation Environment Triggers Biological Responses in Flies) program at LNGS, we sought to elucidate the role of a specific component of the environmental radiation field on the LRE-dependent sensitivity to IR-induced DNA damage responses [12]. Since it is not possible to restore the radiation field present at RRE in LRE, as a first approximation, we decided to increase the dose rate value present at LRE regardless of the type of radiation that generates it. The idea was to place an irradiator in LRE to increase the dose rate value of the gamma component. To this end, we took advantage of especially designed Marinelli beakers consisting of aluminum hollow cylinders filled with tuff (a natural gamma emitter building material) and sealed to avoid any radon exposure due to the tuff decay products [12]. The low-LET component measurements carried out in LRE using TLD-700H placed inside the cylindrical hole of Marinelli filled with tuff (herein irradiator) revealed a dose rate value slightly higher than that of RRE (~100 nGy/h vs. ~66 nGy/h).

We have previously shown that the Marinelli beaker without tuff (phantom) does not affect the growth and fertility of flies grown inside the cylinders at RRE [12], indicating that this device represents an appropriate tool to test the flies’ responses in both LRE and RRE environments. We measured the frequency of IR-induced CBs in larval neuroblasts from wild-type line grown at both LRE and RRE inside the irradiator and the phantom using the challenging dose approach described above. Interestingly, we found that the CB frequency in larvae raised for a complete developmental cycle from embryo to third instar larvae (7 days) inside the irradiator at LRE and then exposed to an acute dose of 10 Gy of gamma rays was ~40% lower than the CBs found in larvae maintained either inside or outside the phantom at LRE (Figure 2b). This result indicates that the partial restoring of low-LET flux (specifically of gamma rays) at LRE is able to rescue the IR sensitivity of flies maintained in a reduced radiation background and is direct evidence of the involvement of the radiation field in the different behavior observed in our biological model system at LRE and RRE. It is reasonable to speculate that there is a threshold value of the dose rate between 20 nGy/h and 66 nGy/h (the reference external value) that switches the response of our biological system, making it pass from one state to another. What we know is that a further increase in the dose rate, obtained by keeping the larvae in Marinelli with tuff at RRE, does not significantly change the radiosensitivity in terms of the CBs induced by 10 Gy (data not shown).

Our results are in agreement with the pioneering work by Planel et al. (1987) on *Paramecium tetraurelia*, showing that the ciliate growth rate decreases inside the Pyrenees Mountain underground laboratory but is restored when cultures are exposed to very low doses of gamma radiation in the same underground environment [26].

More recently, Castillo et al. (2022) carried out an analysis of the effects on the cell number and viability and gene expression of Chinese hamster V79 cells under two background-radiation-deprived conditions (below background and in the presence of KCl source) with respect to the external natural background. Their results indicate that cells growing below background show lower viability than that of KCl-amended underground controls or surface controls. This effect appears after 5 days of incubation and lasts, intermittently, for up to 21 days. The data suggest that the sole emission of γ-rays from K-40 in KCl-amended controls is able to only partially rescue the V79’s viability, and although there is a clear differentiation between the underground and the surface controls, the authors argue that it could depend on differences among the radiation spectrum in the two conditions, although the experimental set up doesn’t allow for the exclusion of the potential influence of other environmental factors such as air pressure or gas composition [5].

## 3. Materials and Methods

### 3.1. Drosophila Strains

The *Drosophila Oregon R (OR-R)* line was used as a wild-type strain. The *Drosophila* mutant alleles *mre*^11^, *nbs*^1^, and *rad50*^Δ5.1^ in genes encoding for DDR factors and *cav*^1^, *moi*^1^, and *eff^tre^*^1^ in telomere capping encoding genes were previously described [27,28,29,30,31]. The detailed information on these stocks as well as images of adult and larval wild-type developmental stages is available on *Flybase* (http://flybase.org (accessed on 11 April 2022)). Flies were maintained in *Drosophila* medium (Nutri-FlyrGF; Genesee Scientific, San Diego, CA, USA) treated with propionic acid at controlled parameters of temperature (22 °C), relative humidity (about 55%rH), and 12 h light-dark alternation in identical cooled incubators (BioloG-Lux140 Cooled Incubator; F.lli Galli G.& P., Fizzonasco, Italy) placed at LRE and RRE. As the Gran Sasso underground laboratory is a tunnel with horizontal access, the difference in atmospheric pressure between the two environments of RRE (approx. 903 mbar) and LRE (approx. 906 mbar) is negligible. To create generation 1, an RRE population of young adults was divided into two groups. One group was kept in RRE and the other transferred to LRE. Both in RRE and LRE, adults were left to lay embryos for 3–4 days and then discarded. Deposited embryos represent generation 0. Young adults from generation 0 were transferred to a new vial for 3–4 days (and then discarded) to lay embryos for generation 1. Young adults from a generation were used as parents for the next generation and so on for the following generations (2–5). The generations 1 and 5 were analyzed to measure the frequency of CBs.

### 3.2. Dosimetry

Radiation dosimetry at RRE and at LRE of the different components of the radiation field was performed with measurements and evaluations from literature data, as previously described [9,12]. Briefly, the neutron dose rate calculated at RRE was 2.5 nGy/h, while at LRE it was considered negligible. The dose rates related to the gamma component were 22 nGy/h at RRE and about 20 nGy/h at LRE. The dose rate due to the directly ionizing component of cosmic rays (mainly muons) is negligible for the underground laboratory LRE, and of about 41 nGy/h at RRE. Overall, at LRE and RRE, we have a dose rate of about 20 nGy/h and 66 nGy/h, respectively. Moreover, measurements of ^222^Rn concentration in the air were obtained using a Radon meter (AlphaGUARD P30, Saphymo Instruments GmbH, Montigny-le-Bretonneux, France). In the present experiments, for both RRE and LRE, the values of Radon concentration were in the range 15–25 Bq/m^3^.

### 3.3. Irradiation Treatments

Third instar larvae were exposed to an acute dose of 10 Gy of gamma rays from a ^137^Cs source at a dose rate of about 0.7 Gy/min using the Gammacell Exactor 40 (Nordion) of the Istituto Superiore di Sanità (ISS), Italy.

To increase the low-LET component of the underground radiation spectrum, we developed an irradiator consisting of an especially designed Marinelli beaker consisting of aluminum hollow cylinders filled with tuff and sealed to avoid any radon exposure due to tuff decay products (see [12]). For the Marinelli irradiator experiment, 10 young males and 10 young females were crossed overnight and then discarded. Laid embryos were kept inside the Marinelli irradiator (with tuff) or in the Marinelli phantom (without tuff) for 7 days in both LRE and RRE.

### 3.4. Chromosome Cytology and Microscopy

Metaphases of colchicine-treated neuroblasts from *Drosophila* third instar larval brains were obtained as previously described [32,33]. For CB scoring, larval brains were dissected 3 h after irradiation and incubated for one additional hour with 10^−6^ M colchicine before preparing the slides. The CBs and TFs were analyzed using the inverted fluorescence microscope Nikon TE 2000 (Nikon Instruments Inc., Melville, NY, USA) equipped with a Charged-Coupled Device (CCD camera; Photometrics CoolSnap HQ). At least 100 metaphases (for mutant strains) and 400 (for *Oregon-R*) for each experimental condition were counted for the statistical analysis.

### 3.5. Statistical Analysis

For experiments with *Oregon-R*, we compared the means of two data sets each obtained from at least three independent experiments. For each experiment, the mean number of CBs per cell was determined (>400 cells for each condition). Error bars show the standard deviation. Having two independent groups of samples (collected independently of one another) that follow normal distributions, we used a parametric test (unpaired Student’s *t*-test) to determine statistical differences between the means obtained from these two different groups. Values of *p* < 0.05 were considered as statistically significant.

For the mutants’ analysis, no statistical differences were observed between LRE and RRE (*t*-test). At least 100 metaphases for each mutant were counted for the statistical analysis.

## 4. Conclusions

Despite the difficulties in eliminating all confounding factors when comparing the underground to the reference environment, our results show that the natural background radiation dose rate is indeed capable of modulating the radiation-induced DNA damage response in the complex multicellular organism *Drosophila melanogaster*, probably through the involvement of stress defense mechanisms. However, additional experiments using different endpoints and/or other biological systems should be carried out to substantiate and possibly extend our conclusions. Furthermore, although our results points to the gamma rays as a relevant component in triggering defense mechanisms, we cannot rule out the possibility that other components (e.g., neutrons) of the environmental radiation spectrum may play a similar role. Given that the modulation of neutrons in underground facilities is not an easy task to accomplish, their contribution to the cellular response to radiation-induced DNA damage remains unclear at the present.

## Figures and Tables

**Figure 1 ijms-23-05472-f001:**
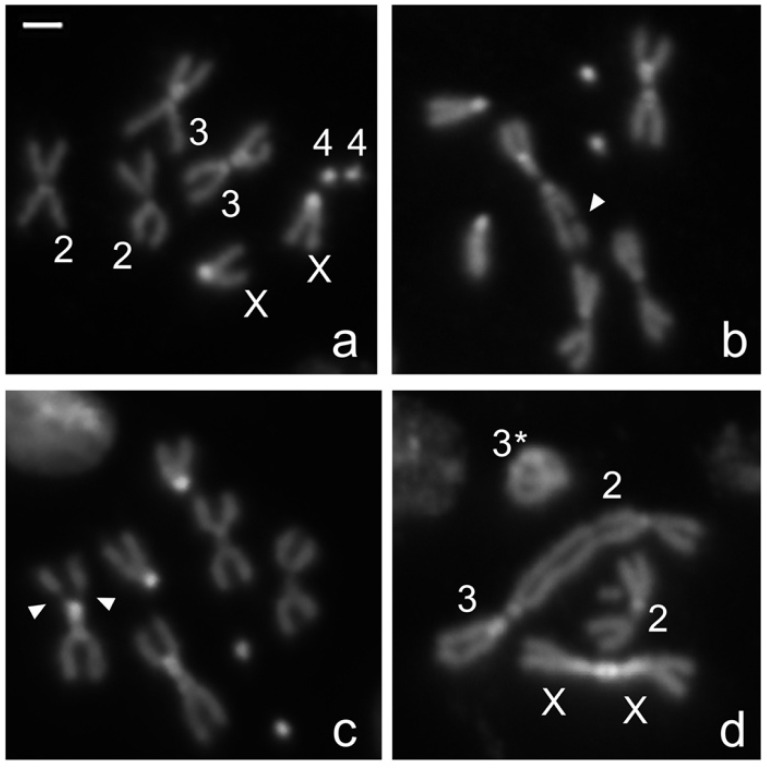
IR-induced CBs and TFs in DAPI-stained larval neuroblasts. (**a**) Examples of IR-induced CBs and TFs in third instar larvae neuroblasts scored for this study: (**a**) wild-type *Drosophila melanogaster* female metaphase, (**b**–**d**) examples of metaphases showing (**b**) autosomal chromatid deletion (arrow), (**c**) autosomal isochromosome deletion (arrows), and (**d**) TFs giving rise to a ring chromosome involving the chromosome 3 (asterisk), a chromosome 2-chromosome 3 dicentric chromosomes, and a X-X dicentric chromosome. Bright DAPI-stained chromosome regions refer to the centromeric and pericentromeric portions of the two major autosomes [23].

**Figure 2 ijms-23-05472-f002:**
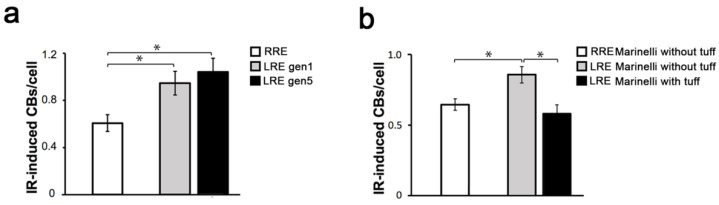
Effects of radiation background on IR-induced CBs. (**a**) Average number of CBs per cell induced by gamma rays in neuroblasts from *Oregon-R* third instar larval brains analyzed after 4 h post-irradiation with the acute dose of 10 Gy. Net data are reported, after subtraction of the control values. CB frequency was analyzed in *Oregon-R* larvae from generations 1 and 5. (**b**). Average number of CBs per cell induced by gamma rays in larvae kept for 7 days in the Marinelli beakers with tuff (irradiator) or without tuff (phantom) at both LRE and RRE and then exposed to the acute dose of 10 Gy. At least three independent experiments were performed for each condition. The error bars represent the standard error of the mean. Values of *p* < 0.05 (*) were considered as statistically significant (*t*-test). In our experiments, we observed a frequency of CBs in the control samples (not irradiated with the acute dose) ranging from 0.03 to 0.15 CBs per cell. CBs: Chromosome Breaks; LRE: Low Radiation Environment; RRE: Reference Radiation Environment.

**Figure 3 ijms-23-05472-f003:**
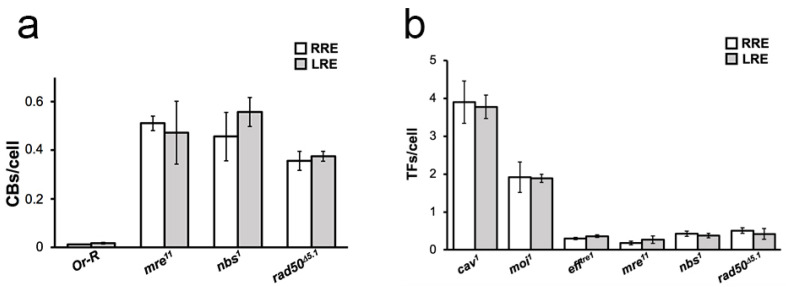
Effects of the deprivation of the radiation background on DDR and telomere capping mutants. (**a**) Analysis of Chromosome Breaks (CBs) in DNA damage response (DDR) mutants. The values refer to the frequency of spontaneous CBs of *Drosophila mre11*, *nbs1*, and *rad50^D5.1^*mutants maintained for five generations and RRE. No statistically significant differences were observed in LRE compared to RRE (*t*-test). (**b**) Analysis of Telomere Fusions (TFs) in telomere capping and DNA damage response (DDR) mutants. The graph shows the frequency of TFs obtained by counting double telomere associations (DTAs) and single telomere associations (STAs) in *Drosophila* strains mutated in genes encoding telomere capping proteins (*cav*^1^, *moi*^1^, and *eff^tre^*^1^) and DNA damage repair factors (*mre*^11^, *nbs*^1^, and *rad50^D^*^5.1^) maintained for five generations at LRE and RRE. No statistically significant differences were observed between LRE and RRE (*t*-test). CBs: Chromosome Breaks; TFs: Telomeric Fusions; LRE: Low Radiation Environment; RRE: Reference Radiation Environment.

## Data Availability

All data generated or analyzed during this study are included in this published article.

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
