# Peer review of "Reduced Environmental Dose Rates Are Responsible for the Increased Susceptibility to Radiation-Induced DNA Damage in Larval Neuroblasts of Drosophila Grown inside the LNGS Underground Laboratory"

_ijms, 2022, doi:10.3390/ijms23105472_

Round 1

Reviewer 1 Report

The study is very interesting and may be very important for understanding the radiation biology, however, several corrections have to be done before it may be considered for publication:

Major:

  • Please move the figures to the results section
  • Add some microscopy photos to show various data used for the statistical analysis and think about showing the photos concerning the morphology of the larvaes and Drosophilas. 
  • Please explain in the text the use of the Student's T-test instead of other statistical methods. 
  • Please discuss the limitations of the study in more detail and limit your conclusions only to the ones from your work. The number of experiments is very little and you should be more careful about such big assumptions as the ones presented in the conclusions. 
  • More detail is needed in the description of the conditions in which control and experimental samples were cultivated.

Minor:

  • Maybe including the data from supplementary would be beneficial to the paper?
  • line 126 "challenging gamma dose of 10Gy; four hours later" before Gy should be a space
  • Text should be formated in MDPI standards - for instance page 1 left side panel should be filled by the Authors 

Otherwise a nice manuscript.

Author Response

The study is very interesting and may be very important for understanding the radiation biology, however, several corrections have to be done before it may be considered for publication:

We are grateful to this reviewer for his/her potential appreciation of our work

  • Please move the figures to the results section

We thank this reviewer for this indication. We have moved the figures on the Result section

  • Add some microscopy photos to show various data used for the statistical analysis and think about showing the photos concerning the morphology of the larvaes and Drosophilas. 

We appreciate the suggestion of this reviewer of providing figures and photos supporting our cytological characterization of our data. To this purpose, we would like to point out to this reviewer that the measurement of frequency of both chromosome aberrations and telomeric fusions has been carried out counting chromosome configurations illustrated in Figure 1b-c and e, respectively. We added in the text the reference to the corresponding aberration indicated in Figure 1. Concerning the request of showing the morphology of Drosophila larvae and adults, we would like to mention that a large amount of images can be indeed found in the Flybase site (flybase.org) that is publically available to the readers. We added a sentence in the Materials and Methods to point this out: “The detailed information on this stock as well as images of adult and larval wild-type developmental stages is available on Flybase (http://flybase.org)”

  • Please explain in the text the use of the Student's T-test instead of other statistical methods

We compare the means of two data sets each obtained from at least three independent experiments. For each experiment the mean of CBs per cell are determined (> 400 cells for each condition). Error bars show the standard deviation. We have two independent groups of samples (collected independently of one another) that follow normal distributions. We use a parametric test (unpaired Student’s t-test) to determine statistical differences between the means obtained from these two groups.

We replaced the previous paragraph of Statistical Analysis in the  Materials and Methods section with the following text about the use of the Student’s t-test: “Having two independent groups of samples (collected independently of one another) that follow normal distributions, we used a parametric test (unpaired Student’s t-test) to determine statistical differences between the means obtained from these two different groups. Values of p<0.05 were considered as statistically significant.’’

  • Please discuss the limitations of the study in more detail and limit your conclusions only to the ones from your work. The number of experiments is very little and you should be more careful about such big assumptions as the ones presented in the conclusions. 

We thank this reviewer for his/her suggestion. We have modified the conclusions as follows: “Despite the difficulties in eliminating all confounding factors when comparing the underground to the reference environment, our results show that the natural background radiation dose rate is indeed capable to modulate the radiation induced DNA damage response in the complex multicellular organism Drosophila melanogaster, probably through the involvement of stress defense mechanisms. However, additional experiments using different endpoints and/or other biological systems should be carried out to substantiate and possibly extend our conclusions. Furthermore, although our results points to the gamma rays as a relevant component in triggering defense mechanisms for this effect, we cannot rule out the possibility that other components (e.g. neutrons) of the environmental radiation spectrum may could play a similar role. Given that the modulation of neutrons in underground facilities is not an easy task to accomplish, their contribution to the cellular response to radiation induced DNA damage remains unclear at the present.”

We also would like to point out that in the Material and Methods sections our indication “At least 100 metaphases for each condition were counted for the statistical analysis” is an underestimation because it mainly refers to telomeric and DDR mutants which elicit strong mitotic proliferation defects (in other words we need to dissect dozens of larvae to obtain a number of cells suitable for our analysis). However, for the experiments performed on wild-type strains (Oregon-R) the number of cells analyzed was much higher (at least 400 cells for each condition). We added this specification in the corresponding part of Materials and Methods: At least 100 metaphases (for mutant strains) and 400 (for Oregon-R) for each experimental condition were counted for the statistical analysis.

  • More detail is needed in the description of the conditions in which control and experimental samples were cultivated

We added some details in Materials and Methods: “Flies were maintained on Drosophila medium (Nutri-FlyrGF; Genesee Scientific) treated with propionic acid at controlled parameters of temperature (22°C), relative humidity (about 55%rH), and 12h light-dark alternation in identical cooled incubators (BioloG-Lux140 Cooled Incubator; F.lli Galli G.& P., Italy) placed at LRE and RRE. As the Gran Sasso underground laboratory is a tunnel with horizontal access, the difference in atmospheric pressure between the two environments RRE (approx. 903 mbar) and LRE (approx. 906 mbar) is negligible.’’

Minor:

  • Maybe including the data from supplementary would be beneficial to the paper?

Thank you for your valuable suggestion. The supplementary data have been moved to the main body of the text.

  • line 126 "challenging gamma dose of 10Gy; four hours later" before Gy should be a space

We fixed this typo.

  • Text should be formated in MDPI standards - for instance page 1 left side panel should be filled by the Authors 

I guess the left panel is filled by MDPI after the submission process is completed. Indeed in our previous articles, this information was added after the article was accepted.

Reviewer 2 Report

Porrazzo et al. have assessed the effects of low level radiation on chromosomal breaks under controlled reduced radiation environment  at the underground lab, the Gran Sasso National Laboratory (LNGS). The authors determined that drosophila melanogaster cells kept under low radiation environment (LRE) at the lab develops higher numbers of chromosomal breaks on acute radiation exposure compared to cells kept at reference radiation environment (RRE). Surprisingly, exposure of cells from LRE to low irradiation rescued the higher chromosomal break phenotype. The authors suggest a stress mediated response to radiation at RET that governs the capacity of those cells to tolerate future high irradiation. Further, the article add to proofs suggesting non-linear response to radiation and one of the first experiments under controlled environment studying low radiation dose effects on genomic instability. This topic is quite relevant for the special issue's reader. 

An issue with the article that needs either discussion in article by the authors or clarification as to why they have not discussed the results or tried to verify result from their extremely relevant recent article in Biorxiv using additional assays (article name: Low dose/dose rate y irradiation protects Drosophila melanogaster chromosomes from double strand breaks and telomere fusions by modulating the expression of Loquacious) (link: https://www.biorxiv.org/content/10.1101/2021.07.23.453515v1.full ).

Author Response

  • Porrazzo et al. have assessed the effects of low level radiation on chromosomal breaks under controlled reduced radiation environment  at the underground lab, the Gran Sasso National Laboratory (LNGS). The authors determined that drosophila melanogaster cells kept under low radiation environment (LRE) at the lab develops higher numbers of chromosomal breaks on acute radiation exposure compared to cells kept at reference radiation environment (RRE). Surprisingly, exposure of cells from LRE to low irradiation rescued the higher chromosomal break phenotype. The authors suggest a stress mediated response to radiation at RET that governs the capacity of those cells to tolerate future high irradiation. Further, the article add to proofs suggesting non-linear response to radiation and one of the first experiments under controlled environment studying low radiation dose effects on genomic instability. This topic is quite relevant for the special issue's reader.

We are grateful to this reviewer for his/her positive comments on our work

  • An issue with the article that needs either discussion in article by the authors or clarification as to why they have not discussed the results or tried to verify result from their extremely relevant recent article in Biorxiv using additional assays (article name: Low dose/dose rate y irradiation protects Drosophila melanogaster chromosomes from double strand breaks and telomere fusions by modulating the expression of Loquacious) (link: https://www.biorxiv.org/content/10.1101/2021.07.23.453515v1.full).

We really appreciate the reviewer enthusiasm on our BiorXiv article. We felt not to discuss our results in line with those of our BiorXiv article mainly because the article is not published yet as we are still working on the reviewers comments. Needless to say that we found this reviewer suggestion very interesting and we are planning to address potential comparisons from these two experimental set-ups in a future review/perspective.

Round 2

Reviewer 1 Report

The authors have revised the paper according to my comments and now I think it may be considered for publication in IJMS.